# Current Evidence Regarding Biomarkers Used to Aid Postoperative Delirium Diagnosis in the Field of Cardiac Surgery—Review

**DOI:** 10.3390/medicina56100493

**Published:** 2020-09-24

**Authors:** Paweł Majewski, Małgorzata Zegan-Barańska, Igor Karolak, Karolina Kaim, Maciej Żukowski, Katarzyna Kotfis

**Affiliations:** 1Department of Cardiac Surgery, Ceynowa Hospital, 84-200 Wejherowo, Poland; lek.majewski.pawel@gmail.com; 2Department of Anesthesiology and Intensive Therapy, Regional Specialist Hospital, 72-300 Gryfice, Poland; 3Department of Anesthesiology, Intensive Therapy and Acute Intoxications, Pomeranian Medical University in Szczecin, 70-111 Szczecin, Poland; karolinakaim@hotmail.com (K.K.); zukowski@pum.edu.pl (M.Ż.); katarzyna.kotfis@pum.edu.pl (K.K.); 4Student Science Club, Department of Anesthesiology, Intensive Therapy and Acute Intoxications, Pomeranian Medical University in Szczecin, 70-111 Szczecin, Poland; igor.karolak@gmail.com

**Keywords:** postoperative delirium, cardiac surgery, biomarkers, perioperative cognitive disorders, neuroinflammation

## Abstract

Postoperative cognitive disorders after cardiac surgery may manifest as postoperative delirium (POD) or later as postoperative cognitive dysfunction (POCD). The incidence of POD after cardiac surgery ranges from 16% to 73%. In contrast to POD, POCD is usually diagnosed after the discharge from hospital, with an incidence of 30 to 70% of cases, very often noticed only by close relative or friends, decreasing after six (20–30%) and twelve (15–25%) months after surgery. Perioperative cognitive disorders are associated with adverse short- and long-term effects, including increased morbidity and mortality. Due to the complexity of delirium pathomechanisms and the difficulties in the diagnosis, researchers have not yet found a clear answer to the question of which patient will be at a higher risk of developing delirium. The risk for POD and POCD in older patients with numerous comorbidities like hypertension, diabetes, and previous ischemic stroke is relatively high, and the predisposing cognitive profile for both conditions is important. The aim of this narrative review was to identify and describe biomarkers used in the diagnosis of delirium after cardiac surgery by presenting a search through studies regarding this subject, which have been published during the last ten years. The authors discussed brain-derived biomarkers, inflammation-related biomarkers, neurotransmitter-based biomarkers, and others. Work based on inflammation-related biomarkers, which are characterized by the low cost of implementation and the effectiveness of delirium diagnosis, seems to be the closest to the goal of discovering an inexpensive and effective marker. Currently, the use of a panel of tests, and not a single biomarker, brings us closer to the discovery of a test, or rather a set of tests ideal for the diagnosis of delirium after cardiac surgery.

## 1. Introduction

Delirium is a neurobehavioral syndrome caused by a transient disruption of normal neuronal activity mediated by alteration in neurotransmitter and neuronal network function, occurring secondary to systemic (metabolic) disturbances [1]. The Diagnostic and Statistical Manual of Mental Disorders 5th Edition (DSM 5) defines delirium as an acutely developing deficit in attention (reduced ability to direct, focus, sustain, and shift attention), coupled with a change in cognition (memory deficit, disorientation, or perceptual disturbance) [2].

Delirium is a strong predictor of worse outcomes and serious complications in hospitalized patients [3]. Similarly, the occurrence of postoperative delirium (POD) is associated with an increased number of different complications, such as prolonged hospitalization, compromised rehabilitation, deterioration of the quality of life, which results in a greater burden on the health care system and increased mortality [4]. Furthermore, patients with postoperative delirium are prone to delay neurocognitive recovery, but also at risk for a five-fold increased chance of nosocomial complications, poor 1-year functional recovery, and even ten-fold increased risk of death after surgery [5].

Despite the publication of DSM-5 criteria for delirium, up to 75% of patients remain undiagnosed as there is a lack of routine monitoring of delirium in many hospital wards [6]. Some of its symptoms may be masked by sedation and fluctuate during the day, which makes recognition of postoperative delirium more difficult [7].

There is a growing interest in the use of objective parameters that may offer a reliable assessment of cerebral damage caused by cardiac surgery, but researchers still look for useful biomarkers, which could help clinicians in their everyday practice to suspect postoperative delirium. Therefore, the detection of elevated or lowered levels of biomarkers, which could serve as predictors or indicators of delirium, seems to be very promising. Moreover, such biomarkers may assist in risk stratification, monitoring of treatment, but primarily may help to diagnose delirium [8].

## 2. Types of Delirium

Delirium can be divided into three subtypes: hyperactive, hypoactive, and mixed [9]. The hypoactive subtype is associated with higher 6-month mortality and a higher mortality rate at 18 months compared to mixed delirium [10]. In the hyperactive subtype, the increased activity levels and speed of action or speech are all associated with involuntary movements, loss of control of activity, restlessness, abnormal content of verbal output, hyperalertness, irritability, and/or combativeness. These patients often require significant involvement of the medical staff due to dangerous and disruptive behavior and receive the most clinical focus in the postoperative period but are also prone to receiving excessive sedation.

In the clinical picture of hypoactive delirium, reduced activity, apathy, decreased amount or speed of speech, decreased alertness, withdrawal, unawareness, or hypersomnolence predominate. This subtype may remain undiagnosed due to the insufficient expression of symptoms. Thereby, patients with hypoactive delirium more often suffer from pressure ulcers and hospital-acquired infections [11]. It should also be underlined that hypoactive delirium is more difficult to diagnose, and the consequences of not treating this condition are definitely worse [10,11,12].

A mixed subtype may be the most common motoric subtype of delirium in intensive care [13]. It is characterized by the variability of the severity and occurrence of symptoms and the difficulty of making an appropriate diagnosis due to the often rapidly changing nature and frequency of the symptoms [14,15].

In the current literature, the term subsyndromal delirium (SSD) has also been proposed when the patient demonstrates cognitive and attention deficits without meeting all the diagnostic criteria for delirium [16]. This condition also leads to an increased length of stay (LOS) in the hospital [17].

Another attempt to classify delirium is based on its etiology and was proposed by Girard et al. This classification includes sedation-associated, sepsis-associated, hypoxic, metabolic, and unclassified delirium subtypes [18]. Moreover, the working group of Evered et al. recommended that the term ‘perioperative neurocognitive disorders’ should be used as an overarching term for cognitive impairment identified during the perioperative or postoperative period and aligned this expression with the terms used in the community. A summary of this approach is depicted in Figure 1 [19].

## 3. Delirium after Cardiac Surgery and Risk Factors

Postoperative delirium has been identified more than a half-century ago in cardiac surgery patients and still remains an unresolved problem [20]. The risk for developing POD and delayed cognitive recovery is associated with advanced age and numerous comorbidities like hypertension, diabetes, depression, and previous ischemic stroke [21,22]. Additionally, the predisposing cognitive profile for neurological conditions has not yet been fully elucidated [23].

The incidence of delirium after cardiac surgery ranges from 16% to 73%, depending on the type of surgery and methodology used for delirium detection. It has been reported that up to 85% of cases remain unrecognized [24]. In the study by Sauer et al., 12.5% of cardiac patients developed delirium during the hospital stay [25]. The authors assessed delirium using the Confusion Assessment Method for the Intensive Care Unit (CAM-ICU) scale during the first four postoperative days and then at 1 month and 1 year after surgery. Data suggested that patients with POD are at higher risk of developing a decline in cognitive performance than those without POD, but this relationship only applies to the first months after surgery and not in the long term [25]. Nevertheless, POD may be treated as an independent risk factor of postoperative cognitive dysfunction [26]. In cardiac surgery, the problem of postoperative delirium usually appears within the first five days after the procedure, and many other confounding factors need to be excluded [27,28]. Delayed neurocognitive recovery is usually revealed at the time of discharge from the hospital (30–70%), very often noticed only by close relatives or friends, and then its incidence decreases after six (20–30%) and twelve (15–25%) months after surgery [29,30,31]. It is generally more frequent after cardiac operations as compared to other types of surgery [32]. Bhamidipati et al. reported that despite significant progress in the postoperative management in cardiac surgery, the percentage of patients experiencing perioperative cognitive disorder had not decreased with more noticeable serious complications, including mortality [33]. This could be due to the fact that, nowadays, more elderly patients undergo cardiac surgery, and this group of patients presents with a variety of comorbidities and systemic vascular disease [34].

The specifics of cardiac surgery, apart from patient-related factors, add further elements that increase the risk of postoperative delirium. These are related to the use of cardiopulmonary bypass (CPB), the complexity of the surgical procedure, including aortic clamping, and the choice between the on-pump and off-pump techniques, which overlap with the occurrence of patient dependent factors. One of the hypotheses for the development of POD is the development of systemic inflammatory response syndrome (SIRS) not only due to the cardiac surgery itself but also due to the exposure of the patient to the adverse effect of CPB [35]. However, several randomized studies have established that CPB does not affect the incidence of POD, and the use of CPB has no effect on the level of inflammatory markers [36]. Numerous studies have also confirmed that off-pump surgery does not reduce the incidence of POD [37,38,39,40]. Part of the risk factors, which contribute to the occurrence of postoperative delirium after cardiac surgery, may be modified in the real-world setting; some of them can be regulated only partially, but still, a few remain out of the reach of the clinicians (Table 1).

## 4. Pathomechanisms of Delirium after Cardiac Surgery

So far, the mechanisms underlying postcardiac delirium have not been fully elucidated. Many different theories and multifactorial processes describe its development, and neuroinflammation is one of them [41]. The injury of the blood-brain barrier (BBB) and inflammatory response after cardiac surgery are described in human and animal investigations [42,43,44,45]. One of the theories explaining the emergence of delirium is the systems integration failure hypothesis (SIFH), which assumes that non-modifiable factors overlap modifiable elements at the same time. This can lead to the decompensation of the “system” that manifests itself as delirium [46,47]. In the SIFH concept, delirium is supposed to consist of a combination of specific neurotransmitter secretion disorders with inappropriate processing of sensory information and motor response [47] and is also a multifactorial process involving mechanisms, such as neuroinflammation and oxidative stress [48,49].

The important element determining the occurrence of delirium is a disruption in the integrity of the BBB, which consists of the presumed central neuroinflammatory response and which is a consequence of launching multiple potential pathways under the influence of peripheral or systemic factors [1]. The imbalance of biomarkers, including tumor necrosis factor-alpha (TNF-α), interleukin (IL)-1, IL-6, IL-8, and brain susceptibility to neuronal injury, as well as the activation of microglia via neuroinflammation, may be the result of the exposure to cardiopulmonary bypass, which contributes to neuroinflammation [50]. Androsova et al. and Rudolph et al. mentioned that systemic inflammatory response syndrome is the key factor, which may affect the brain, and lead to neuroinflammation with IL-6 and TNF-α as biomarkers in predicting delirium [51,52].

There are additional studies, which underline the role of the intraoperative cerebral desaturation, oxygen delivery and utilization, embolic load, hypo- and hyperthermia, or intraoperative glucose homeostasis [53], also potential neurotoxicity of anesthetic dosage [54,55,56], but other invariably emphasize the fact that the inflammatory response plays a key role in the pathogenesis of POD (Figure 2) [33,35,57].

## 5. Diagnosing Delirium after Cardiac Surgery

Structured assessment and monitoring of postoperative delirium after surgery and anesthesia should be performed in accordance with the European Society of Anesthesiology POD guidelines [27]. Numerous other evidence-based guidelines advocate the monitoring of delirium as a part of routine clinical care. Nevertheless, it still has not entered into standard clinical practice in many places, and further work is required to improve the routine delirium monitoring [19,24,58].

Performing specific and sensitive neuropsychological tests and examinations that focus on different cognitive domains [5] is time-consuming, exhausting, and require engagement from trained experts. Despite that, their sensitivity is still insufficient [27]. Additionally, tests are not free of interfering confounders, such as ceiling or floor (basement) effect. Glumac et al. found over 100 different neuropsychological tests that were being used for the diagnosis of delirium [59]. Difficulties and problematic use of them result indirectly from the non-uniform use of the definition of POD and postoperative cognitive dysfunction (POCD) [19,58]. The comparison of data is, therefore, additionally difficult. Due to the multitude of diagnostic tests, many studies cannot be compared, and the results obtained in many different ways may mask the real number of patients with delirium [19]. Sheth et al. proposed a period of seven days from the time of cardiac surgery until the test for cognitive impairment due to the direct influence of drugs and the hospital environment on the patient’s condition, while Glumac et al. stated that the tests should be performed preoperatively and in the early postoperative period (seven to 14 days after the procedure), but also even 6 to 9 months after surgery [59,60].

This abundance of information and diagnostic options for delirium assessment has led to the issuing of the guidelines that recommend routine use of validated tools. European Society of Anesthesiology (ESA) recommends using a validated delirium score for postoperative delirium screening (for example, CAM-ICU and Nursing Delirium Screening Scale (NuDesc) [27].

## 6. The Aim of the Narrative Review

The aim of this narrative review was to identify and describe biomarkers used in the diagnosis of delirium after cardiac surgery, which have been presented in papers published between 2010 and 2020. The authors used PubMed open resource database for this purpose and rejected studies conducted in children and on animals. Specific keywords used were biomarkers, delirium, cardiac surgery, postoperative delirium, POCD, and neuroinflammation.

## 7. Biomarkers in Delirium Diagnosis

Reliable and rapid assessment of delirium after cardiac surgery is of paramount importance. Biomarker-based diagnostics take on special significance and the benefits of a faster diagnosis of delirium in patients at risk. In some circumstances, this may be lifesaving. The usage of biomarkers as factors, facilitating the diagnosis and monitoring the delirium course, would allow to increase the diagnostic effectiveness, implement early treatment, and reduce the percentage of complications-associated delirium. The ideal delirium marker should be detected earlier than delirium occurrence, should be highly sensitive, correlate with the severity of the disease, stable, translational, easy to obtain, independent of physiological variables, low-cost, readily available, and have high validity and specificity to detect delirium. Moreover, it should be associated with a known mechanism (localized damage). The search for an ideal delirium marker is a complex process as many confounding factors account for delirious state postoperatively [61]. It is much more probable that this will be based on the identification of a panel of biomarkers in order to make an accurate and quick diagnosis [51,62].

There is a growing interest in the use of objective parameters that may offer a correct assessment of cerebral damage caused by cardiac surgery. Biomarkers of the central nervous system (CNS) injury arise after neurons and glial cells damage produced during cardiac surgery; this, in turn, causes active or passive intracellular components release outside CNS. There are two mechanisms in which we are able to detect the biomarkers of injury of the CNS. First, the local biomarker concentration increases in the CNS, and it diffuses into the circulation through the BBB, and the second involves damage of the BBB with unregulated penetration of the biomarker into the bloodstream [63]. Furthermore, the overexposure to cytokines observed in the excessive activation of the inflammatory cascade of the central nervous system may lead to oligodendroglial neurotoxicity, potentially contributing to apoptosis and demyelination [64].

Studies regarding biomarkers in delirium in different clinical conditions laid the groundwork for further, more recent research, and a new pathway for detecting delirium after cardiac surgery. To enable clarity, we divided the biomarkers of delirium after cardiac surgery into the following categories: brain-derived biomarkers, inflammation-related biomarkers, neurotransmitter-based biomarkers, and others. 

## 8. Brain-Derived Biomarkers

During acute illness, the permeability of the BBB changes. One of the markers of its disruption and the injury of the astrocytes after CNS insult is calcium-binding protein S100 beta (S100B) [65,66,67]. The S100B is mostly secreted by astrocytes under conditions of metabolic stress. Blyth et al. identified that circulating S100B concentrations accurately reflect BBB dysfunction [66]. S100B has a short half-life and is considered to be a concentration-dependent biomarker, the high levels of which result from active excretion of the protein or passive secretion by damaged tissues [68]. The result is the change in synaptic transmission, neural excitability, and cerebral blood flow, leading to the neurobehavioral and cognitive symptoms characteristic of delirium [50]. It has been considered as a putative biomarker of CNS damage, and increased cerebrospinal fluid (CSF) and serum levels are connected with the adverse CNS outcomes, especially delirium [69,70,71]. Westaby et al. showed a good correlation between higher S100B concentrations a few hours after the surgery with postoperative cognitive decline [72]. Both active and passive secretion plays a role in the elevation of the S100B in the CNS injury [72]. In the process of neuroinflammation, biomarker S100B may play a role as a validated measure of BBB disruption and contribute to delirium [67]. In delirious patients, high S100β levels in CSF and serum have been linked to adverse CNS outcomes [69,70,71,72].

As reported by Fazio et al. and Einav et al. in cardiac surgery, S100B elevation has turned out to be dependent on patient-related factors like age, sex, hypertension, as well as perioperative factors, including the use of cell saver, the presence or absence of extracorporeal pump, the degree of perfusion during and after the cardiopulmonary bypass [73,74]. Many researcher groups (in the past) have analyzed serum levels of S100 in patients undergoing cardiac surgery to assess brain dysfunction that occurs as a result of the surgical procedure; however, the relationship between S100B levels and neurological and neurophysiological findings remains not well defined. Together with the insulin-like growth factor-1 (IGF-1), the S100B was proposed by Khan et al. as a promising and specific marker for delirium [61].

Tau protein is a microtubule-associated protein involved in stabilizing the axonal cytoskeleton and vesicles transport in the neuronal synapse. Tau hyperphosphorylation is associated with neuronal death observed in neurodegenerative disorders, such as Alzheimer’s disease (AD), and is a well-known marker associated with this disease [75,76]. In the study by Palotas et al. [77], increased Tau levels and reduced-amyloid peptide in CSF were observed in the presence of cognitive impairment in patients who had undergone cardiopulmonary revascularization. These data suggest certain parallelism between changes in biomarkers present in cognitive decline that occurs after cardiac surgery and Alzheimer’s disease. Reinsfelt et al. showed high levels of Tau after aortic valve replacement in 10 patients. Tau elevation was noticed in the cerebrospinal fluid within 24 h before and after cardiac surgery and strongly connected with the degree of microembolization intraoperatively [45]. In the study by Saller at al., a multi-fold increase of preoperative levels of tau protein, neurofilament (NfL), and glial fibrillary acidic protein (GFAP) was observed. However, research was done on only nine cases, with different time-points of discharge. Nevertheless, the authors concluded that both Tau and NfL might be of benefit to identify patients in cardiac surgery at risk for delirium and to detect patients with the postoperative emergence of delirium [78].

It is interesting that Simons et al. also found that ICU patients with hypoactive delirium were susceptible to differentiation from those without delirium by measurement of Tau protein. The authors showed significantly higher levels of serum Tau protein and the Tau/Aβ1–42 ratio in hypoactive patients. In the same study, they also found higher levels of neopterin and IL-10 in hypoactive patients than in the mixed-type delirium group [79].

GFAP is produced by glial cells and astrocytes. Following CNS injury and during neurodegeneration, GFAP gene activation and protein induction appear to play a critical role in the astroglial cell activation (astrogliosis). When apoptosis occurs, GFAP leaks into the CSF and blood, and its breakdown products are released into biofluids, making them strong candidate biomarkers for such neurological disorders [80]. Gailiušas et al. demonstrated a significant increase in GFAP level during the perioperative period and found a significant correlation not only with delirium occurrence but also with the lowest mean arterial pressure (MAP) during surgery [81]. Contrary, Saller et al. claimed that GFAP levels seemed to be the least suitable for delirium prediction, detection, and monitoring [78].

Ubiquitin carboxyl-terminal hydrolase L1 (UCH-L1) is neuron-specific and necessary for normal synaptic and cognitive function [82]. Wang et al. reported that the UCH-L1 gene might be linked to AD, traumatic brain injury, and Parkinson’s disease, and the increased level of UCH-L1 in the extracellular fluid suggests neuronal injury [83]. DiMeglio et al. showed a persistent elevation of neurodegeneration markers among patients undergoing cardiac surgery, including the UCH-L level, which lasted up to 3 months after surgery [84]. It has been recognized as a neuron-specific enzyme and marker for direct neuron injury in traumatic brain injury patients [85], and a link between UCH-L1 and neurodegeneration has been suggested [86]. Its role in delirium as a biomarker, which directly detects injury to neurons, could provide a pathway to predict and screen patients at risk for delirium following cardiac surgery [41]. Up to date, there are no studies on the predictive value of UCH-L1 in the perioperative period.

Together with S100B protein, neuron-specific enolase (NSE) is one of the most commonly used biomarkers of cerebral injury for the assessment of neurological disorders. Yuan stated that S100B and NSE could be reliable indicators, reflecting the cerebral damages secondary to cardiac surgery [87]. Enolases are found as dimers resulting from the association of three distinct subunits (α, β, and γ), and the α-subunit is expressed in most tissues. NSE is passively released into the bloodstream by cell destruction, and its elevation in the late phases of injury may be explained by delayed cell death. NSE is found in the peripheral blood only in negligible amounts, and it is found in platelets and erythrocytes, particularly after hemolysis, and, therefore, hemolysis may cause a falsely positive NSE elevation [88]. Besides, increased NSE levels are observed after CPB along with S100B [89,90]; it is also increased in patients with neurological complications after acute myocardial infarction [91], cardiac arrest with resuscitation, and profound hypothermia [92]. Although older studies have shown an increase of NSE and S100B serum levels and neurocognitive dysfunction in cardiac surgery [93,94], research regarding delirium after cardiac surgery still remains an open issue [95].

Research regarding UCL-H1, brain-derived neurotrophic factor (BDNF), and metalloproteases-9 (MMP-9) has so far focused on brain injury following cardiac surgery. MMP-9 increase has been reported in patients undergoing cardiac surgery with extracorporeal circulation [96]. Due to the pathomechanism of delirium development and their possible relationship with the development of delirium in patients after cardiac surgery, these biomarkers constitute an interesting direction for further research. 

Brain-Derived Biomarkers are summarized in Table 2.

## 9. Inflammation-Related Biomarkers

Nemeth et al. studied C-reactive protein (CRP) and procalcitonin (PCT) levels for the evaluation of the inflammatory response role in the pathogenesis of POCD [31]. For the purpose of their study, the authors defined PCT > 0.5 μg/L and CRP > 5 mg/L as elevated on the first postoperative day after cardiac surgery. The separation into the low inflammatory response (LIR, PCT ≤ 0.5 μg/L) and high inflammatory response (HIR, PCT > 0.5 μg/L) was based on PCT levels only. Neurological tests performed in the study were mini-mental state examination, trail making tests A and B, digit symbol test, Stroop color, and word test [31]. In the group of 42 patients enrolled in the study, the LIR and HIR groups differed significantly in the PCT, but not regarding the CRP. The authors conclude that the direct relationship between the magnitude of non-infective inflammatory response and incidence of POCD has not been established. This is consistent with the data reported by Çinar et al. that CRP, although elevated, cannot be considered as an independent risk factor in the development of POD [98]. On the other hand, a study, including the CRP levels performed by Kotfis et al., showed a correlation between increased CRP levels and delirium [62].

Kazmierski et al. showed an elevation of IL-2 and TNF-α levels in the postoperative period in coronary artery bypass graft (CABG) patients with delirium, and these results were independent of patient-related factors like age and gender, as well as their psychiatric and physical state and co-morbidities or perioperative factors like CPB time or duration of surgery [99]. According to Çinar et al., it is difficult to consider the postoperative value of TNF-α or CRP as an independent risk factor for POD [98]. However, Androsova et al. mentioned that SIRS is the key factor, which may affect the brain and lead to neuroinflammation with IL-6 and TNF-α as biomarkers in predicting delirium [51]. Rudolph et al. stated that different chemokines, like IL-4 and IL-10, were elevated after cardiac surgery in patients who developed delirium in the early postoperative period [52]. In addition, Plaschke et al. found that early postoperative delirium after cardiac surgery was characterized by increased Il-6 and plasma cortisol levels, accompanying changes within the bilateral bispectral index analysis [100].

Elevated inflammatory biomarkers have been indicated as predictors of dementia and delirium in the general population [101]. Inflammation-related biomarkers are summarized in Table 3.

## 10. White Cells-Derived Biomarkers

Neutrophils and leukocytes are important factors in the development of inflammation [102]. The increased neutrophil-to-lymphocyte ratio (NLR) has been studied in patients after procedures with CPB, including CABG and valve procedures. Theologou et al. showed that the risk of developing POD was associated with prolonged endotracheal intubation and prolonged ICU stay, along with peaked NLR, urea, creatinine, and sodium levels [103]. This finding might suggest that an inadequate reaction of the immune system may play a role in the pathogenesis of delirium. This has led research towards cheap and easily available markers, such as platelet-to-lymphocyte ratio (PLR), as well as a platelet-to-WBC ratio (PWR).

In a retrospective cohort analysis on a total group of 1904 patients, Kotfis et al. studied NLR, PLR, and PWR with CRP for the establishment of an easy-to-obtain predictor of delirium in cardiac surgery patients [62]. The study showed that in the delirium group, which was over 13% of the total study population, most of the delirious patients were older men with comorbidities, such as hypertension, extracardiac atherosclerosis, and chronic renal failure, and this group experienced a long time of postoperative mechanical ventilation with significantly higher mortality both at 30 days and 1 year after surgery. Kotfis et al. showed that lower preoperative mean PLR values and lower PWR values along with higher mean levels of leukocytes and CRP were found in patients with POD. In patients undergoing planned cardiac surgery, the PWR occurred to be a novel and independent predictor of postoperative delirium. The calculation of white cells-derived biomarkers (NLR, PLR, PWR) requires no extra cost and could be an important factor for the recognition of delirium after cardiac surgery.

Still, further research is needed to describe the relationship between the PWR and delirium. Yet, an inadequate immune system response could play a major role in POD in cardiac surgery as it has been shown by the white blood cells differential count and the CRP increase in delirium patients. In order to obtain an inexpensive, less time-consuming, and possible to use in everyday clinical practice marker of delirium, Kotfis et al. used laboratory data together with an objective pre-operative examination to produce a CARDEL index, which is based on the function of age, glycosylated hemoglobin (HbA1c), and PWR. The analysis showed that the CARDEL index was more accurate in predicting the development of delirium after CABG than any of these factors alone [62]. White cells-derived biomarkers are summarized in Table 3.

## 11. Neurotransmitter-Based and Other Biomarkers

Inflammation-related changes in the level of neurotransmitters (deficiency of acetylcholine in the brain) and impaired cholinergic transmission have been described as a potential mechanism for delirium development [104]. However, the postoperative measurement of acetylcholinesterase (AChE) or butyrylcholinesterase (BChE) does not distinguish patients with and without POD [105]. The influence of CPB and blood product transfusion on AChE and BChE needs further studies. Moreover, postoperative treatment with AChE inhibitor (rivastigmine) is unsuccessful in preventing POD after cardiac surgery [106].

Neopterin is produced by monocytes and macrophages. It can be found in plasma, urine, and cerebrospinal fluid in patients with delirium and is produced due to the traumatic and systemic factors in the development of SIRS [107]. Osse et al. found that in older patients (aged 70 or more) undergoing cardiac surgery, high preoperative levels of neopterin predicted a higher prevalence of delirium [108]. Moreover, postoperative neopterin and homovanillic acid (HVA) levels were found to be associated with POD [108].

In humans, cortisol is one of the stress hormones that, if chronically elevated, has a major influence on the brain by causing damage to the hippocampus and impairing hippocampus-depended learning and memory. Its secretion is proportional and positively correlated with the severity of surgical stimuli [109]. In a study by Mu et al., elevated serum cortisol on the first day after surgery was highly correlated with an increased risk of POD, with prolonged postoperative ICU and hospital stay [110]. In a similar study, Kazmierski et al. investigated the impact of increased preoperative and postoperative cortisol concentration in relation to the risk of developing postoperative delirium in patients with preoperative cognitive impairment [111].

Insulin-like growth factor (IGF-1) has been reported to be involved in the pathophysiology of delirium. The role of IGF-1 is neuroprotective as it inhibits the cytotoxic cytokines (e.g., TNF-α). In a study by Çinar et al., the authors found that low levels of IGF-1 before the cardiac operation were risk factors for developing POD [98]. Neurotransmitter-based and other biomarkers are summarized in Table 4. 

## 12. Discussion

In this narrative review, the authors searched for new data regarding biomarkers used in the detection of postoperative delirium in the last years. Delirium has been constantly a challenging clinical condition. Biomarkers used as an additional element in the diagnostic pathway may contribute to an increased detection rate of delirium and translate into a better quality of postoperative care, also in cardiac surgery departments. Furthermore, the use of biomarkers as a factor, facilitating the diagnosis and monitoring the delirium course, would allow to increase the diagnostic effectiveness and reduce the percentage of complications resulting from an insufficient diagnosis of delirium, which is associated with longer hospitalization, higher treatment costs, and increased mortality [112,113]. A useful delirium biomarker should be easily identifiable, readily available, and enable targeted therapy while being cost-effective.

Research regarding biomarkers of delirium after cardiac surgery has been carried out for several years, as has research on other biomarkers of delirium in other clinical conditions. Due to the complexity of the mechanisms of delirium development and the difficulties in diagnosis, researchers have not yet found a clear answer to the question of which patient will be at a higher risk of developing delirium [114]. Of note, no biomarker has been identified that would enable the development of treatment strategies to lower the incidence, severity, or duration of POD. When it comes to a pathophysiological explanation based on CNS injury during CPB, the neurotransmitters hypothesis was described a long time ago, but none of them brings clinicians closer to recognizing delirium before the symptoms occur. It is mostly due to the diversity of biochemical substances used as test factors, as well as the non-standardized way of monitoring and assessing delirium in the study groups. These are serious elements, limiting the drawing of conclusions and not allowing to indicate a specific marker as the most appropriate in the assessment of delirium after cardiac surgery [19]. Besides, inflammatory biomarkers do not bring promising results, although they are easily available and relatively inexpensive. The researchers have found that inflammatory biomarkers are not consistently reported as delirium risk factors. A study based on the white cells-derived markers (NLR, PLR, and PWR) seems promising, and thanks to their effectiveness and the ease of obtaining the results, they may become a helpful factor in the diagnosis and monitoring of delirium after cardiac surgery [61]. It has also been proposed to use NLR as a potential marker for delirium prediction in patients with other medical problems (i.e., acute ischemic stroke) [115].

On the one hand, established theories, such as the role of neurotransmitters, inflammation, and stress response in delirium pathogenesis, give researchers the ground to search for an ideal biomarker. On the other hand, most of them lack sufficient evidence or validation for implementation in clinical practice and are not readily available.

The majority of the studies suggest that biomarkers could be useful in assessing the delirium risk. Brain-derived biomarkers have an established place in determining the severity of CNS injury and outcomes, but when it comes to delirium, it is not so obvious. This is one of the reasons why researchers should use consistent study methodology to classify inflammatory processes and other independent mechanisms underlying the development of POD. Due to a variety of symptoms and differences in the subtypes of delirium, the diagnosis may be difficult to establish, and delirium often remains unrecognized [3]. It is particularly difficult in patients from cardiac surgery departments who are admitted to the hospital in a serious condition, who are often elderly, with numerous comorbidities, and sometimes even presenting with underlying cognitive disorders. It is worth mentioning that the diagnostic pathway with the use of biomarkers may be especially useful when treating patients with hypoactive delirium, as these patients may often go unnoticed in the postoperative department. At the moment, it seems reasonable to conclude that only a properly selected panel of biomarkers can be used to help diagnose POD.

Furthermore, it seems unclear whether changes in the levels of biomarkers are associated with or play a causative role in POD after cardiac surgery, or whether they merely represent an acute-phase response to cardiac surgery and CPB. This requires further research. But most importantly, it should be remembered by the clinical teams that effective delirium management is based on early detection of delirium, and the treatment is mainly non-pharmacological. It relies upon the cooperation of different members of the medical, nursing, and physiotherapy teams, as well as family members. It should include effective pain assessment and treatment restoration of senses (glasses, hearing aids), early mobility [97,116,117], and effective coordinated postoperative management to limit complications and the length of hospitalization [118,119,120,121].

## 13. Conclusions

Delirium after cardiac surgery is a common clinical problem that is associated with longer hospital stay and may be characterized by an increased frequency of complications, including increased mortality. So far, no ideal delirium marker has been found for patients after cardiac surgery. Nevertheless, previous work based on inflammation-related biomarkers, which are characterized by the low cost of implementation and the effectiveness of delirium diagnosis, seems to be the closest to the goal of discovering an inexpensive and effective marker. Currently, the use of a panel of tests, and not a single biomarker, brings us closer to the discovery of a test, or rather a set of markers ideal for aiding early diagnosis of delirium after cardiac surgery. This requires further confirmation in clinical trials.

## Figures and Tables

**Figure 1 medicina-56-00493-f001:**
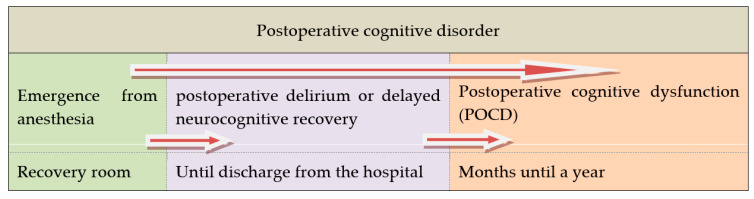
Postoperative cognitive disorder in a time frame (after Evered et al. [19]).

**Figure 2 medicina-56-00493-f002:**
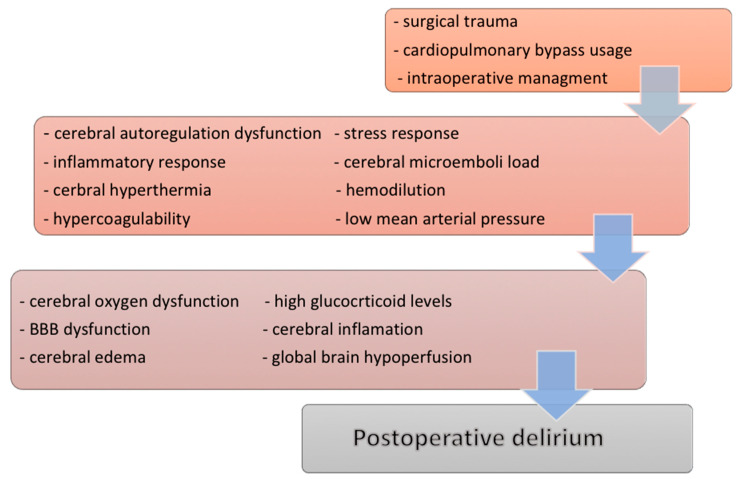
Diagram of the possible mechanisms in the development of the postoperative cognitive disorder in cardiac surgery. Legend: BBB—blood-brain barrier.

**Table 1 medicina-56-00493-t001:** Modifiable and non-modifiable risk factors involved in the emergence of perioperative cognitive disorder.

	Modifiable	Partially Modifiable	Nonmodifiable
**Preoperative**	Glycemic control Substance abuse Pre-rehabilitation Blood pressure control	Depression Neurocognitive reserve Depression and anxiety Social adjustment Education	Age Genetics Dementia Neurodegenerative disease Underlying vascular disease (atherosclerosis) Other organs disease (kidney, liver, pulmonary)
**Intraoperative**	Surgical technique Cardiopulmonary bypass Temperature management Blood pressure management Glycemia, ion, pH management	Anesthetic management and brain monitoring (e.g., BIS, cerebral oximetry) Duration of surgery Operation technique (on-pump vs. off-pump) Type of incision (median sternotomy vs. lateral thoracotomy)	Type of surgery Direct myocardial injury
**Postoperative**	Duration of mechanical ventilation Sedation, analgesia, and delirium (ABCDEF approach)	Postoperative complication–early management Patient frailty Sleep disturbance	Hospital environment

Legend: ABCDEF: A—Assess, prevent, and manage pain; B—Both spontaneous awakening trials (SATs) and spontaneous breathing trials (SBTs); C—Choice of analgesia and sedation; D—Delirium: Assess, prevent, and manage; E—Early mobility and exercise; F—Family engagement and empowerment. BIS—Bispectral index.

**Table 2 medicina-56-00493-t002:** Brain-derived biomarkers reported in research on post-cardiac surgery delirium.

Biomarker	Localization/Function	Research First Author (Ref.)	Cardiac Surgery Research Main Findings and Other Main Comments
**S100B**	astroglial, Schwann cells	Salameh [36], Steinmetz [97]	CPB does not affect the incidence of POD and POCD, and the use of CPB also has no effect on the S100B biomarker
		Fazio [73], Einav [74]	S100B elevation has turned out to be dependent on patient-related and peri-operative factors
**NSE**	neural tissue, neuroendocrine cells	Gao [89], Gu [90]	Increased levels observed after CPB, no research in post cardiac surgery delirium
		Gailiušas [81]	An increase in NSE level during the perioperative period may be associated with subclinical neuronal damage
**NfL**	neurons, cytoskeleton	Saller [78]	Increased level postoperatively, might be of benefit in the detection of POD
**GFAP**	astroglial cytoskeleton	Gailiušas [81]	Increase in GFAP observed after CPB
		Saller [78]	Increased level, the least suitable for POD prediction
**Tau protein**	axonal cytoskeleton	Reinsfelt [45]	High levels of Tau after AVR
		Palotas [77]	High Tau levels in patients after cardiopulmonary revascularization
		Saller [78]	Increased Tau protein level postoperatively, nine cases, different time of discharge
**MMP-9**	cerebral blood vessels	Sokal [96]	Studies on SIRS after cardiac surgery, no research in post cardiac surgery delirium in adults
**UCL-H1**	neurons and neuroendocrine cells	DiMeglio [84]	Persistent elevation after cardiac surgery

Legend: AVR aortic valve replacement, CPB cardiopulmonary bypass, GFAP glial fibrillary acidic protein, ICU intensive care unit, MMP metalloproteases, NfL neurofilament, NSE neuron-specific enzyme, POD postoperative delirium, POCD postoperative cognitive dysfunction, UCL-H1 ubiquitin C terminal hydroxylase-L1, SIRS systemic inflammatory response syndrome.

**Table 3 medicina-56-00493-t003:** Inflammation-related and white cells-derived biomarkers reported in research regarding POD in cardiac surgery.

Biomarker	Localization	Research First Author (Ref.)	Cardiac Surgery Research Main Findings and Other Main Comments
**IL-2**	Serum	Kazmierski [99]	Elevation in postoperative period after CABG
**IL-6**	Serum	Plaschke [100]	Increased stress levels and inflammatory reaction correlate with early postoperative delirium after cardiac surgery
**Chemokines (IL-4, IL-10)**	Serum	Rudolph [52]	After cardiac surgery, chemokine levels are elevated in subjects who develop delirium in the early postoperative period
**TNF-alpha**	Serum	Kazmierski [99]	Elevation in postoperative period after CABG
		Çinar [98]	Increased postoperatively, no differences between delirious and non-delirious patients
**PCT**	Serum	Nemeth [31]	No relation between non-infective inflammatory response and the incidence of POCD
**CRP**	Serum	Nemeth [31]	No relation between non-infective inflammatory response and the incidence of POCD
		Kotfis [62]	Higher CRP postoperatively
		Çinar [98]	Increased postoperatively, no differences between delirious and non-delirious patients
**NLR** **PLR** **PWR**	Serum	Theologou [103]	High NL ratio after procedures with total CPB, incl. AVR + MVR ± CABG
		Kotfis [62]	Lower PLR, PWR preoperatively Higher leukocytes postoperatively Development of the CARDEL Index

Legend: AVR aortic valve replacement, CABG coronary artery bypass graft, CPB cardiopulmonary bypass, CRP C-reactive protein, IL interleukin, MVR mitral valve replacement, NLR neutrophil-to-lymphocyte ratio, PCT procalcitonin, PLR platelet-to-lymphocyte ratio, POCD postoperative cognitive decline, PWR platelet-to-WBC ratio, TNF-α tumor necrosis factor-alpha.

**Table 4 medicina-56-00493-t004:** Neurotransmitter-based and other biomarkers reported in research on post-cardiac surgery delirium.

Biomarker	Localization/Function	Research First Author [Reference]	Cardiac Surgery Research Main Findings and Other Main Comments
**IGF-1**	Serum/growth factor	Çinar [98]	Low pre-operative levels are detected in the delirium group
**AChE** **BChE**	Serum/neurotransmitter	John [105]	No evidence in the diagnosis of POD after cardiac surgical procedures
**Neopterin**	Serum/pteridines	Osse [108]	Preoperative high levels of neopterin
**HVA**	Serum /amino acid	Osse [108]	Increased postoperative level associated with POD
**Cortisol**	Serum/hormone	Kazmierski [111]	High plasma levels in delirious patients

Legend: AChE Acetylcholinesterase, BChE Butyrylcholinesterase, HVA homovanillic acid, ICU intensive care unit, IGF-1 Insulin-like growth factor, POD postoperative delirium.

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
