# Peer review of "Current Evidence Regarding Biomarkers Used to Aid Postoperative Delirium Diagnosis in the Field of Cardiac Surgery—Review"

_medicina, 2020, doi:10.3390/medicina56100493_

Round 1

Reviewer 1 Report

Major:

  • Authors should adhere to the wording of Evered et al (citation 24 of the current reference list). While the wording suggested by Evered ('perioperative neurocognitive disorders') is correctly cited on page 3 of the manuscript, authors use the word ‘acute brain dysfunction’ and later on ‘POCD’. This starts in the abstract and continues throughout the article. Here it would be preferable is the authors briefly clarify this aspect at the beginning of their introduction, introduce the correct wording and use it consequently throughout the text body and the tables.
  • Table 1 and several explanations give the misleading idea that there is a continuum between emergence delirium, POD and later POCD. While some patients will develop POD after emergence delirium, others will not. POD is not a prerequisite for developing POCD. This might be clarified graphically (e.g. arrows directly going to POD or POCD, while other arrows lead to POCD through ED and POD). It seems that the authors stick to this continuum throughout the rest of the article (Figure 1: the last bubble is named ‘Postoperative delirium and POCD’). However, much of the later presented evidence is on POD only. It would be much easier for the reader if the authors clarify the distinction between POD and POCD while explaining the recommendation by Evered et al and then focus the rest of the article on POD. If the material allows, the article might be split into two articles, one of biomarker for acute events (POD) and one for delayed neurocognitive recovery respectively postoperative neurocognitive disorders.
  • Furthermore, the distinction between POD and ICU delirium seems arbitrary. Depending on capacity, patients with identical severity of disease will be transferred to PACU, normal ward while in other hospitals; they might be directly transferred to ICU. If the authors insist on the 5-days post-OP-criterion for POD, they should provide a reference for this. The ESA-guideline of the prevention of POD (ref # 67 in the current manuscript) uses a 5-days criterion to define the onset of POD.
  • The whole introduction which does not deal with biomarker (this is what the reader expects to read after the title) should be shortened and summarized in on paragraph no longer than 10-15 lines. Diagnosis and risk factors in this article are only relevant concerning their impact on biomarkers. Or in other words: the reader expects explanations on how the later described biomarker reflect ‘pathomechanism of delirium’, i.e. underlying biological processes leading to POD. Whether CAM-ICU or NuDesc are appropriate screening tools for POD does not play any importance in an article on biomarker for POD.

Minor:

  • Authors use the wording ‘categories’, ‘form’ and ‘subtype’ when they talk about ‘types of delirium’. This should be collapsed to one consistent wording, e.g. ‘subtype’.
  • It is not correct that the ESA recommended CAM-ICU or NuDesc for POD delirium detection. (as stated in page 6, first two lines). In fact, the ESA guideline states (page 198, right column) that for the recovery room Nu-DESC and CAM / CAM-ICU have been validated against DSM criteria. However, their specific use is not recommended. The recommendation in Table 6 is: ‘We recommend using a validated delirium score for POD screening’.  – Not a single specific screening tool.
  • Unclear what the authors mean by ‘perfect delirium marker’ (page 6, second paragraph, line 6 from top). High validity, high specificity? Low costs? This should be explained at this point of the article. This likewise concerns the title of the whole manuscript. Please consider rewording. The rewording should inform the reader that the article is a narrative review.
  • When it comes to the biomarker, several studies have not been cited by the authors:
    • S100B: the study my Herrmann et al. is missing (doi: 10.1016/s1010-7940(99)00245-6.)
    • Tau and neurofilament: the case series of Thamas Saller and colleagues was not considered and cited by the authors (doi: 10.5507/bp.2019.043.)
    • The authors use the wording “Glial fibrillatory acidic protein (GFAP)”, Table 3. However, according the PubMed, the correct wording seems to be “Glial fibrillary acidic protein”. Nevertheless, a study from the working group of Szwed and colleagues from Bydgoszcz on glial fibrillatory/fibrillary acidic protein, neuroserpin, phosphorylated axonal neurofilament subunit H, and visinin-like protein 1 in patients undergoing Coronary Artery Bypass Grafting is missing (doi: 10.1016/j.athoracsur.2019.10.071.)
    • Neuron-specific enolase (NSE): the authors did not mention the study of Gailiušas  et al in elective CABG patients (doi: 10.6001/actamedica.v26i1.3949.).
    • Authors do mention briefly the role of ‘intraoperative glucose homeostasis’ (page 5, line 6 from top). It might be interesting for the reader to get more information on this ‘simple biomarker’, which is routinely monitored in the perioperative setting. See e.g. the article by Zhou et al. on impaired fasting glucose on brain injury during cario-surgery (doi: 10.1080/08941939.2018.1519049.).

  • Authors should add the reference numbers in tables 4-6 in order that the reader may easily find the references.

Author Response

  1. Reviewer

Comments and suggestion

Authors replies

Major

       Authors should adhere to the wording of Evered et al (citation 24 of the current reference list). While the wording suggested by Evered ('perioperative neurocognitive disorders') is correctly cited on page 3 of the manuscript, authors use the word ‘acute brain dysfunction’ and later on ‘POCD’. This starts in the abstract and continues throughout the article. Here it would be preferable is the authors briefly clarify this aspect at the beginning of their introduction, introduce the correct wording and use it consequently throughout the text body and the tables.

 I could not agree more with the reviewer that majority of the publications about postoperative delirium lack consistence in terms of time, relationship with delayed changes etc. It seems that Evered et al. nicely describe the nomenclature and it should be widely adapted.

Changes were made according to the reviewer’s suggestion.

The aim of this manuscript is not in clarifying the definition of perioperative neurocognitive disorders. In this publication, the authors analyzed available sources to retrieve data regarding different biomarkers.

·         Table 1 and several explanations give the misleading idea that there is a continuum between emergence delirium, POD and later POCD. While some patients will develop POD after emergence delirium, others will not. POD is not a prerequisite for developing POCD. This might be clarified graphically (e.g. arrows directly going to POD or POCD, while other arrows lead to POCD through ED and POD). It seems that the authors stick to this continuum throughout the rest of the article (Figure 1: the last bubble is named ‘Postoperative delirium and POCD’). However, much of the later presented evidence is on POD only. It would be much easier for the reader if the authors clarify the distinction between POD and POCD while explaining the recommendation by Evered et al and then focus the rest of the article on POD. If the material allows, the article might be split into two articles, one of biomarker for acute events (POD) and one for delayed neurocognitive recovery respectively postoperative neurocognitive disorders.

Changes were made according to the reviewer’s remarks.

It is true that this manuscript deals only with postoperative delirium.

Amendments in the table and figure were done.

·         Furthermore, the distinction between POD and ICU delirium seems arbitrary. Depending on capacity, patients with identical severity of disease will be transferred to PACU, normal ward while in other hospitals; they might be directly transferred to ICU. If the authors insist on the 5-days post-OP-criterion for POD, they should provide a reference for this. The ESA-guideline of the prevention of POD (ref # 67 in the current manuscript) uses a 5-days criterion to define the onset of POD.

The authors agree.

We changed the text accordingly.

·         The whole introduction which does not deal with biomarker (this is what the reader expects to read after the title) should be shortened and summarized in on paragraph no longer than 10-15 lines. Diagnosis and risk factors in this article are only relevant concerning their impact on biomarkers. Or in other words: the reader expects explanations on how the later described biomarker reflect ‘pathomechanism of delirium’, i.e. underlying biological processes leading to POD. Whether CAM-ICU or NuDesc are appropriate screening tools for POD does not play any importance in an article on biomarker for POD.

This has been changed.

Minor:

·         Authors use the wording ‘categories’, ‘form’ and ‘subtype’ when they talk about ‘types of delirium’. This should be collapsed to one consistent wording, e.g. ‘subtype’.

This has been changed.

·         It is not correct that the ESA recommended CAM-ICU or NuDesc for POD delirium detection. (as stated in page 6, first two lines). In fact, the ESA guideline states (page 198, right column) that for the recovery room Nu-DESC and CAM / CAM-ICU have been validated against DSM criteria. However, their specific use is not recommended. The recommendation in Table 6 is: ‘We recommend using a validated delirium score for POD screening’.  – Not a single specific screening tool.

This has been changed.

·         Unclear what the authors mean by ‘perfect delirium marker’ (page 6, second paragraph, line 6 from top). High validity, high specificity? Low costs? This should be explained at this point of the article. This likewise concerns the title of the whole manuscript. Please consider rewording. The rewording should inform the reader that the article is a narrative review.

The text has been added.

The title has been amended.

·         When it comes to the biomarker, several studies have not been cited by the authors:

·         S100B: the study my Herrmann et al. is missing (doi: 10.1016/s1010-7940(99)00245-6.)

·         Tau and neurofilament: the case series of Thamas Saller and colleagues was not considered and cited by the authors (doi: 10.5507/bp.2019.043.)

·         The authors use the wording “Glial fibrillatory acidic protein (GFAP)”, Table 3. However, according the PubMed, the correct wording seems to be “Glial fibrillary acidic protein”. Nevertheless, a study from the working group of Szwed and colleagues from Bydgoszcz on glial fibrillatory/fibrillary acidic protein, neuroserpin, phosphorylated axonal neurofilament subunit H, and visinin-like protein 1 in patients undergoing Coronary Artery Bypass Grafting is missing (doi: 10.1016/j.athoracsur.2019.10.071.)

·         Neuron-specific enolase (NSE): the authors did not mention the study of Gailiušas  et al in elective CABG patients (doi: 10.6001/actamedica.v26i1.3949.).

·         The authors tried to include only the most recent studies, based on publications from the last decade.

·         Has been added.

·         Was released in August 2020. Off-pump research

·         Reconsidered, after reading whole manuscript – scarce description of methods and data, we decided not to include.  

·         Authors do mention briefly the role of ‘intraoperative glucose homeostasis’ (page 5, line 6 from top). It might be interesting for the reader to get more information on this ‘simple biomarker’, which is routinely monitored in the perioperative setting. See e.g. the article by Zhou et al. on impaired fasting glucose on brain injury during cario-surgery (doi: 10.1080/08941939.2018.1519049).

·          

Citation added.

·         Authors should add the reference numbers in tables 4-6 in order that the reader may easily find the references.

·          

This has been done.

Reviewer 2 Report

Where are we in the challenge of finding the perfect biomarker to aid delirium diagnosis after cardiac surgery?

I read with great interest the mentioned manuscript. The general aim of this review was to identify and describe biomarkers used in the diagnosis of delirium after cardiac surgery by presenting a search through studies in cardiac surgery which have been published during the last ten years. The topic is of some interest. Overall, the manuscript needs copyediting to assist the reader. However, the manuscript suffers from some shortcomings. My specific comments are below:

Major comments

Introduction

  • Page 3; Line 35: It was not clear to me the difference of POD to early POCD.

Inflammation-related biomarkers

  • Page 9; Inflammation-related biomarkers, Paragraph 1: you cited the CRP, but did not described no study correlating their dosage to delirium after CABG.

White-cells derived biomarkers

  • Page 9; 1 paragraph: These cited studies of biomarkers are not related to delirium and CABG

Tables

  • Table 3 is not necessary

Minor comments

1) Page 2 Line 13: Include REF

2) Page 2 Line 15: Include REF

3) Page 3; Line 41: Include REF

4) Page 2 Line 30: The aim of the study must be insert just before the description of the biomarkers. All the text before is part of the Introduction.

5) Page 5; Line 5: Review the sentence: “embolic load, hypo-“

6) Page 5; Line 8: POCD > and POD?

7) Page 5; Line 20: Include REF

8) Page 5; Line 26: Review: “ceiling, floor (basement) effect”

9) Page 5; Line 27: Include REF

10) Page 5; Line 28: POCD > and POD?

11) Page 5; Line 37: Define SCCM before

12)Page 6: missing space before next topic > “Biomarkers in delirium diagnosis”

13) Page 6; Line 1: Define ESA and NuDesc before

14) Page 6; Topic (Brain-derived biomarkers); Line 3: Review “calcium-binding protein S100 beta (S100B)”. Its in Negrito

15) Page 6; Topic (Brain-derived biomarkers); Line 14: Define CSF before

16) Page 7; 3 paragraph; Line 1: Review “Tau protein”. It’s in Negrito

17) Page 7; 3 paragraph; Line 3: Define Alzheimer’s disease in the first appearance

18) Page 8; Line 1: Review “Glial fibrillary acidic protein (GFAP)”. It’s in Negrito

19) Page 8; 2 paragraph: GFAP previously named

20) Page 8; 3 paragraph: Review “Ubiquitin carboxyl-terminal hydrolase L1 (UCH-L1)”. It’s in Negrito

 21) Page 8; 3 paragraph: Defined PD

22) Page 9; 1 paragraph: Review “neuron-specific enolase (NSE)”. It’s in Negrito

23) Page 9; 2 paragraph: Review “Brain-derived neurotrophic factor”. It’s in Negrito

24) Page 9; 3 paragraph: Review “C-reactive protein (CRP) and procalcitonin (PCT)”. It’s in Negrito

25) Page 9; 4 paragraph: Review “IL-2 and TNF-α and IL-4 and IL-10”. It’s in Negrito

26) Page 9; 5 paragraph: Defined NL in the first appearance

27) Page 9; 1 paragraph: Review “IL-6”. It’s in Negrito

28) Page 9; 1 paragraph: CRP previously named

29) Page 10; 1 paragraph: Review “neutrophil-to-lymphocyte ratio (NLR); platelet-to-lymphocyte ratio (PLR), and platelet-to-WBC ratio (PWR)”. It’s in Negrito

30) Page 11; 1 paragraph: Review “of delirium. Kotfis”

31) Page 11; 1 paragraph: Defined HbA1c in the first appearance

31) Page 11; 3 paragraph: Review “Neopterin”. It’s in Negrito

32) References: it is not according to the Journal standard

Tables and Figures

Table1: it’s really a Figure > review

Table 2: Missing legend

Figure 1: Missing legend

Table 3: Title in another page

Table 3:  I suggest putting at the end of the topic

Table 3: Define PD, PAMP

Table 3: Review “It has a short – life (30-60)” > How much time?

Table 4: Missing the number of the reference associated with the research main authors

Table 4: Missing space after the table

Table 5: Missing the number of the reference associated with the research main authors

Table 6: Missing the description of the cardiac surgery research main findings

and other main comments by Osse

Author Response

2 Reviewer

Comments and suggestion

Authors replies

Major

Introduction

·         Page 3; Line 35: It was not clear to me the difference of POD to early POCD.

Ultimately deleted as there is no hard data to transition from one form to another, furthermore explanation is out of the scope of this manuscript.

Inflammation-related biomarkers

·         Page 9; Inflammation-related biomarkers, Paragraph 1: you cited the CRP, but did not described no study correlating their dosage to delirium after CABG.

Ref. 40 and 74 deal both with PCT and CRP, but we were unable to find publication studied CRP biomarker alone in the context of connection with delirium.

White-cells derived biomarkers

·         Page 9; 1 paragraph: These cited studies of biomarkers are not related to delirium and CABG

Crossed

Tables

·         Table 3 is not necessary

Deleted

Minor comments

1) Page 2 Line 13: Include REF

Has been added, paragraph changed place in the text.

2) Page 2 Line 15: Include REF

Has been added, paragraph changed place in the text.

3)Page 3; Line 41: Include REF

Has been added.

4) Page 2 Line 30: The aim of the study must be insert just before the description of the biomarkers. All the text before is part of the Introduction.

Has been moved and title added

5) Page 5; Line 5: Review the sentence: “embolic load, hypo-“

Has been review and changed.

6) Page 5; Line 8: POCD > and POD?

Right. Has been chaned

7)  Page 5; Line 20: Include REF

Has been added.

8) Page 5; Line 26: Review: “ceiling, floor (basement) effect”

9) Page 5; Line 27: Include REF

Has been added.

10) Page 5; Line 28: POCD > and POD?

Has been changed and reference added.

11) Page 5; Line 37: Define SCCM before

Has been defined.

12)Page 6: missing space before next topic > “Biomarkers in delirium diagnosis”

Has been changed.

13) Page 6; Line 1: Define ESA and NuDesc before

Has been defined.

14) Page 6; Topic (Brain-derived biomarkers); Line 3: Review “calcium-binding protein S100 beta (S100B)”. Its in Negrito

Has been changed.

15) Page 6; Topic (Brain-derived biomarkers); Line 14: Define CSF before

Has been defined.

16) Page 7; 3 paragraph; Line 1: Review “Tau protein”. It’s in Negrito

Has been defined.

17) Page 7; 3 paragraph; Line 3: Define Alzheimer’s disease in the first appearance

Has been defined.

18) Page 8; Line 1: Review “Glial fibrillary acidic protein (GFAP)”. It’s in Negrito

Has been changed.

19) Page 8; 2 paragraph: GFAP previously named

Has been changed.

20) Page 8; 3 paragraph: Review “Ubiquitin carboxyl-terminal hydrolase L1 (UCH-L1)”. It’s in Negrito

Has been changed.

 21) Page 8; 3 paragraph: Defined PD

22) Page 9; 1 paragraph: Review “neuron-specific enolase (NSE)”. It’s in Negrito

Has been changed.

23) Page 9; 2 paragraph: Review “Brain-derived neurotrophic factor”. It’s in Negrito

Has been changed.

24) Page 9; 3 paragraph: Review “C-reactive protein (CRP) and procalcitonin (PCT)”. It’s in Negrito

Has been changed.

25) Page 9; 4 paragraph: Review “IL-2 and TNF-α and IL-4 and IL-10”. It’s in Negrito

Has been changed.

26) Page 9; 5 paragraph: Defined NL in the first appearance

Has been defined.

27) Page 9; 1 paragraph: Review “IL-6”. It’s in Negrito

Has been changed.

28) Page 9; 1 paragraph: CRP previously named

Has been removed

29) Page 10; 1 paragraph: Review “neutrophil-to-lymphocyte ratio (NLR); platelet-to-lymphocyte ratio (PLR), and platelet-to-WBC ratio (PWR)”. It’s in Negrito

Has been changed.

30) Page 11; 1 paragraph: Review “of delirium. Kotfis”

Has been changed.

31) Page 11; 1 paragraph: Defined HbA1c in the first appearance

Has been changed.

31) Page 11; 3 paragraph: Review “Neopterin”. It’s in Negrito

Has been changed.

32) References: it is not according to the Journal standard

Has been changed.

Tables and Figures

Table1: it’s really a Figure > review

Has been changed.

Table 2: Missing legend

Has been added.

Figure 1: Missing legend

Has been added.

Table 3: Title in another page

Has been removed according to major comment.

Table 3:  I suggest putting at the end of the topic

Table 3: Define PD, PAMP

Table 3: Review “It has a short – life (30-60)” > How much time?

Table 4: Missing the number of the reference associated with the research main authors

Has been added.

Table 4: Missing space after the table

Has been added.

Table 5: Missing the number of the reference associated with the research main authors

Has been added.

Table 6: Missing the description of the cardiac surgery research main findings

Has been added.

Round 2

Reviewer 1 Report

  • On page 8 of the revised manuscript, the authors state ‘The aim of this review was to identify and describe biomarkers used in the diagnosis of delirium after cardiac surgery by presenting a systematic search through studies /…/’. This sentence gives the (wrong?) idea that the references included in this review derive from a systematic search and thus fulfil the criteria of a systematic review. If the authors performed a systematic literature search, their search criteria, the inclusion and exclusion criteria need to be explicitly described. The reader would want to see a flow chart, showing the raw list of hits, the number of excluded references and so forth. The reviewer assumes that this is not a systematic review. Hence, the sentence needs correction.
  • If authors state that the search covers references from 2010-2020 [which is, by the way, not a decade, but 11 years], they do not need to include studies before 2010, such as Herrmann et al. (Stroke 2000; 31:645–650), or Kilminster et al. (Stroke. 1999;30(9):1869-1874).
  • The title remains unclear. It does not make much difference to change the adjective ‘perfect’ to ‘ideal’. What is the difference between a perfect and an ideal biomarker? The article is not about the challenges of finding any biomarkers but a narrative review of current evidence on biomarker in the field of cardiac surgery.

Author Response

Reviewer 1

We really appreciate all  valuable comments in order to improve our manuscript.

1.On page 8 of the revised manuscript, the authors state ‘The aim of this review was to identify and describe biomarkers used in the diagnosis of delirium after cardiac surgery by presenting a systematic search through studies /…/’. This sentence gives the (wrong?) idea that the references included in this review derive from a systematic search and thus fulfil the criteria of a systematic review. If the authors performed a systematic literature search, their search criteria, the inclusion and exclusion criteria need to be explicitly described. The reader would want to see a flow chart, showing the raw list of hits, the number of excluded references and so forth. The reviewer assumes that this is not a systematic review. Hence, the sentence needs correction.

Has been changed

  1. If authors state that the search covers references from 2010-2020 [which is, by the way, not a decade, but 11 years], they do not need to include studies before 2010, such as Herrmann et al. (Stroke 2000; 31:645–650), or Kilminster et al. (Stroke. 1999;30(9):1869-1874).

 Has been deleted.

  1. The title remains unclear. It does not make much difference to change the adjective ‘perfect’ to ‘ideal’. What is the difference between a perfect and an ideal biomarker? The article is not about the challenges of finding any biomarkers but a narrative review of current evidence on biomarker in the field of cardiac surgery.

Has been changed : “Current evidence regarding biomarkers used to aid postoperative delirium diagnosis in the field of cardiac surgery – narrative review.

Reviewer 2 Report

I read with great interest the R2 of the mentioned manuscript. However, the manuscript still suffers from some shortcomings. My specific comments are below:

Major comments

Figure 2

  • Figure 2 is repeated

Brain-derived biomarkers

  • Page 8 and 9: The fist paragraph of the chapter is very similar to the second one. Review.

Inflammation-related biomarkers

  • “Nemeth et al. studied C-reactive protein and procalcitonin (PCT) and levels for the evaluation of the inflammatory response role in the pathogenesis of POCD”. In the description of the article, you did not cite purpose of the CRP.

White-cells derived biomarkers

  • The second and third paragraphs are confusing, mixing systemic inflammatory biomarkers with biomarkers derived from white cells.
  • These 2 paragraphs are so repetitive.
  • Page 14; 1 paragraph: You mix again white cell biomarkers and systemic inflammatory biomarkers. 

Neurotransmitter-based and other biomarkers

  • Page 15; 1 paragraph: “However postoperative acetylcholinesterase (AChE) treatment was unsuccessful in distinguishing delirium or cognitive dysfunction after cardiac surgery [118]. Postoperative measurement of AChE yielded no evidence in the diagnosis of POD and the influence of cardiac surgical procedure on AChE needs further studies [119].”. > review this statement, as treatment with AChE does not really help in distinguishing between POD or cognitive dysfunction after cardiac surgery but is has been evaluated as a supposed treatment for preventing delirium in various patient profiles in literature.

Minor comments

1) Figure 1: Place the title before the table

2) Table 1: Place the title before the table and the subtitle after the table

3) Figure 2: Separate the subtitle from the figure title

4) Page10; 2 paragraph: “Glial fibrillary acidic protein” and “central nervous system” were defined before as GFAP and CNS. Change.

5) Page11; 1 paragraph: “cerebrospinal fluid” was defined before as CSF. Change.

6) Page11; 1 paragraph: “Following central nervous system injuries and neurodegeneration, GFAP gene activation…”

7) Page11; 3 paragraph: “Alzheimer Disease” was defined before as AD. Change.

8) Page11; 3 paragraph: “UCH-L1 gene is linked to ADAlzheimer Disease, traumatic brain injury, PDParkinson’s disease, and an increase in extracellular fluid suggesting neuronal injury”. That phrase needs some copy-editing.

 9) Page12; 2 paragraph: “BDNF (Brain-derived neurotrophic factor)” > change the order.

10) Page12; 2 paragraph: “UCH-L1 and NSE were described as markers for direct injury in traumatic brain injury patients. Although Grandi et al. showed that BDNF (Brain-derived neurotrophic factor) levels and NSE γ-subunit of enolase levels were higher in ICU patients who became delirious and other studies showed an increase of NSE serum levels and neurocognitive dysfunction in cardiac surgery [99,100], research on post cardiac surgery delirium remains an open issue [115101]”. That phrase needs some copy-editing.

11) Page 12; Inflammation-related biomarkers: “CRP” was not defined before.

12) Page 12; Inflammation-related biomarkers: “Nemeth et al. studied C-reactive protein and procalcitonin (PCT) and levels for the evaluation of the inflammatory response role in the pathogenesis of POCD”.

13) Page 12; White cells derived biomarkers: “coronary artery bypass grafting” was defined before as CABG.

14) Page 12; White cells derived biomarkers: “The results showed higher risk of developing delirium on the next day after the procedure but also in those who experienced prolonged hospitalization with endotracheal tube and prolonged ICU Stay [106]. Calculation of the NL ratio in the postoperative period takes requires no extra cost and could be a strong factor for the recognition delirium of after cardiac surgery”. That phrase needs some copy-editing.

15) Page 13; 1 paragraph: “IL-6” is in negrito.

16) Page 13; 1 paragraph: “C-reactive protein (CRP)” was defined before.

17) Page 13; 1 paragraph: “neutrophil-to-lymphocyte ratio” was defined before.

18) Page 16; line 1: “postoperative delirium and long-term cognitive dysfunction” was defined before.

Author Response

Reviewer 2

We really appreciate all  valuable comments in order to improve our manuscript.

I read with great interest the R2 of the mentioned manuscript. However, the manuscript still suffers from some shortcomings. My specificcommentsarebelow:

Major comments

Figure 2

  • Figure 2 is repeated – They are repeated partially only, or rather they overlap.

The authors insist on having both Table 1 (risk factors) and Figure 2 (possible mechanisms) for clarity.

Brain-derivedbiomarkers

  • Page 8 and 9: The first paragraph of the chapter is very similar to the second one. Review.

Has been reviewed

Inflammation-relatedbiomarkers

  • “Nemeth et al. studied C-reactive protein and procalcitonin (PCT) and levels for the evaluation of the inflammatory response role in the pathogenesis of POCD”. In the description of the article, you did not cite purpose of the CRP.

Has been chaned.

White-cellsderivedbiomarkers

  • The second and third paragraphs are confusing, mixing systemic inflammatory biomarkers with biomarkers derived from white cells.

Has been changed

  • These 2 paragraphs are so repetitive.

Has been reviewed and changed

  • Page 14; 1 paragraph: You mix again white cell biomarkers and systemic inflammatory biomarkers.

Has been reviewed and changed.

Neurotransmitter-based and Rother biomarkers

  • Page 15; 1 paragraph: “However postoperative acetylcholinesterase (AChE) treatment was unsuccessful in distinguishing delirium or cognitive dysfunction after cardiac surgery [118]. Postoperative measurement of AChE yielded no evidence in the diagnosis of POD and the influence of cardiac surgical procedure on AChE needs further studies [119].”. > review this statement, as treatment with AChE does not really help in distinguishing between POD or cognitive dysfunction after cardiac surgery but is has been evaluated as a supposed treatment for preventing delirium in various patient profiles in literature.

Has been changed ; This has been changed to: “Inflammation-related changes in the level of neurotransmitters (deficiency of acetylcholine in the brain) and impaired cholinergic transmission has been described as a potential mechanism for delirium development [116,117]. However, postoperative measurement of acetylcholinesterase (AChE) or butyrylcholinesterase (BChE) did not distinguish patients with and without POD [118]. The influence of CPB and blood product transfusion on AChE and BChE needs further studies [118]. Moreover, postoperative treatment with AChE inhibitor (rivastigmine) was unsuccessful in preventing POD after cardiac surgery [119].

Minor comments

1) Figure 1: Place the title before the table - DONE

2) Table 1: Place the title before the table and the subtitle after the table - DONE

3) Figure 2: Separate the subtitle from the figure title - DONE

4) Page10; 2 paragraph: “Glial fibrillary acidic protein” and “central nervous system” were defined before as GFAP and CNS. Change. DONE

5) Page11; 1 paragraph: “cerebrospinal fluid” was defined before as CSF. Change. - DONE

6) Page11; 1 paragraph: “Following central nervous system injuries and neurodegeneration, GFAP gene activation…” DONE

7) Page11; 3 paragraph: “Alzheimer Disease” was defined before as AD. Change. DONE

8) Page11; 3 paragraph: “UCH-L1 gene is linked to ADAlzheimer Disease, traumatic brain injury, PDParkinson’s disease, and an increase in extracellular fluid suggesting neuronal injury”. That phrase needs some copy-editing.

DONE. We rephrased this to “Wang et al. have reported that the UCH-L1 gene may be linked to Alzheimer’s Disease, traumatic brain injury and Parkinson’s disease and the increased level of UCH-L1 in the extracellular fluid suggests neuronal injury”

 9) Page12; 2 paragraph: “BDNF (Brain-derived neurotrophic factor)” > change the order. DONE

10) Page12; 2 paragraph: “UCH-L1 and NSE were described as markers for direct injury in traumatic brain injury patients. Although Grandi et al. showed that BDNF (Brain-derived neurotrophic factor) levels and NSE γ-subunit of enolase levels were higher in ICU patients who became delirious and other studies showed an increase of NSE serum levels and neurocognitive dysfunction in cardiac surgery [99,100], research on post cardiac surgery delirium remains an open issue [115101]”. That phrase needs some copy-editing.

DONE This has been changed to: Although Grandi et al. showed that the levels of Brain-derived neurotrophic factor (BDNF) and the γ-subunit of NSE were higher in ICU patients who became delirious [99]. Although other studies also showed an increase of NSE serum levels and neurocognitive dysfunction in cardiac surgery [100, 101], research regarding delirium after cardiac surgery still remains an open issue .

11) Page 12; Inflammation-related biomarkers: “CRP” was not defined before. DONE

12) Page 12; Inflammation-related biomarkers: “Nemeth et al. studied C-reactive protein and procalcitonin (PCT) and levels for the evaluation of the inflammatory response role in the pathogenesis of POCD”. DONE

13) Page 12; White cells derived biomarkers: “coronary artery bypass grafting” was defined before as CABG. DONE

14) Page 12; White cells derived biomarkers: “The results showed higher risk of developing delirium on the next day after the procedure but also in those who experienced prolonged hospitalization with endotracheal tube and prolonged ICU Stay [106]. Calculation of the NL ratio in the postoperative period takes requires no extra cost and could be a strong factor for the recognition delirium of after cardiac surgery”. That phrase needs some copy-editing. DONE

15) Page 13; 1 paragraph: “IL-6” is in negrito. We cannot find this, but the sentence has been changed to: “Both elevated IL-6 and CRP level have been indicated as predictors of cognitive decline and dementia in the general population [107-109]”.DONE

16) Page 13; 1 paragraph: “C-reactive protein (CRP)” was defined before. DONE

17) Page 13; 1 paragraph: “neutrophil-to-lymphocyte ratio” was defined before. DONE

18) Page 16; line 1: “postoperative delirium and long-term cognitive dysfunction” was defined before. DONE
